# Response of Long-Tailed Duck (*Clangula hyemalis*) to the Change in the Main Prey Availability in Its Baltic Wintering Ground

**DOI:** 10.3390/ani12030355

**Published:** 2022-02-01

**Authors:** Paola Forni, Julius Morkūnas, Darius Daunys

**Affiliations:** Marine Research Institute, Klaipėda University, 92294 Klaipėda, Lithuania; juliusmorkunas@gmail.com (J.M.); darius.daunys@ku.lt (D.D.)

**Keywords:** long-tailed duck, *Clangula hyemalis*, sea duck, diet changes, benthic macrofauna, feeding ecology, Baltic Sea, body condition, feeding selectivity

## Abstract

**Simple Summary:**

Extreme change in the benthic macrofauna community after invasion of the bottom fish round goby (*Neogobius melanostomus*) in the eastern Baltic Sea, challenged the declining long-tailed duck (*Clangula hyemalis*) population at the wintering ground. The aim of this study was to assess the change in the diet of the long-tailed duck in two different seabed types: highly impacted hard-bottom and adjacent pristine soft-bottom habitats. The results showed a significant change in the diet in hard-bottom habitats, while the diet in soft-bottom habitats remained similar over time. At the same time, the body condition did not differ over time or between habitats, confirming the high foraging flexibility and the quick species response to the extreme changes in prey composition and availability.

**Abstract:**

The long-tailed duck (*Clangula hyemalis*) is a vulnerable and declining species wintering in the Baltic Sea. The introduction of the invasive fish, the round goby (*Neogobius melanostomus*), dramatically impacted the benthic macrofauna in hard-bottom habitats, while no significant changes occurred in soft-bottom benthic macrofauna. Therefore, we aimed to assess the extent to which the diet of long-tailed ducks changed in two different bottom types. We analysed the stomach content of 251 long-tailed ducks bycaught in gillnets from 2016 to 2020 in hard- and soft-bottom habitats and compared these results with those published by Žydelis and Ruškyte (2005). The results show that the long-tailed duck experienced a change in diet in hard-bottom habitats, shifting from the blue mussel to *Hediste diversicolor*, barnacles, and fish. In soft-bottom habitats, their diet remained similar over time and was based on *H. diversicolor*, a few bivalve species, and *Saduria entomon*. There was no evidence of significant differences in diet between sex or age. Despite the abovementioned changes in diet, the average body condition of the species did not change over time or between habitats. This confirms that long-tailed ducks have high feeding flexibility and quick species response to changes in prey availability, as they are capable of shifting their diet to new prey.

## 1. Introduction

The Baltic Sea is an important site for wintering ducks and other seabirds who migrate from their reproductive grounds in the Artic tundra. The most important areas are the offshore banks and shallow coastal waters, where ducks can easily reach their prey items on the seabed [1].

In the Baltic Sea, many threats are currently affecting the population status of long-tailed ducks. These include bycatch in gillnets, and their habitat is undergoing modification, and ship traffic, oil pollution, hunting, hazardous chemicals, diseases, nutritional deficiencies, and ecosystem changes, including global warming effects, encompassing other threats that this species faces [1,2,3,4]. These threats are the main cause of the long-tailed duck’s population decline, which has decreased by up to 65.4% [2,5]. The evaluation of the distribution and abundance of the long-tailed duck (*Clangula hyemalis*) in the Baltic Sea was assessed twice. The first survey took place from 1992 to 1993 and estimated a population of 4.3 million long-tailed ducks wintering in the Baltic Sea [2], while repeated counts in the same areas from 2007 to 2009 recorded a vertiginous population decrease to 1.5 million. This corresponded to a rapid population decline [2,3], in which the abovementioned threat of long-tailed ducks being bycaught in gillnets was found to be responsible for a 1–5% annual decline alone [6]. For these reasons, the long-tailed duck is classified as a ‘Vulnerable’ species on the IUCN global red list of threatened species [7,8].

In the Baltic Sea, it has been shown that long-tailed ducks can dive up to depths of 20–30 m [9], but some records show that they can dive up to depths of 60 m [10,11]. The species is a generalist and opportunistic feeder with high energetic demands [12]; therefore, they forage on a wide range of food resources [13], mostly on benthic macrofauna (e.g., crustaceans, small bivalves, polychaetes, and gastropods), fish, and even herring eggs [14,15,16]. Individuals dive and select the most profitable prey in terms of feeding costs and energy gain [15,17,18,19], and this finding has been supported by earlier studies that demonstrated a strong species preference for the easily accessible and abundant blue mussel (*Mytilus edulis trossulus* [2,4,15]).

Sea ducks can switch their diet to new prey or move to a new feeding ground if there is a change in the benthic macrofauna community [12,20,21,22,23,24,25]. The flexibility in their diet allows them to maintain the required energy balance and the body reserves that are necessary for wintering survival and reproduction [20]. For example, common scoters (*Melanitta nigra*) changed their distribution pattern in the German North Sea according to the distribution of newly introduced species due to their higher profitability, i.e., higher net energy gained per unit time [17,25,26]. In the Bering Sea, the consequent replacement of the most eaten clam with a new clam species caused spectacled eiders (*Somateria fischeri*) to switch their prey [20]. This ability to shift diet preferences was also studied in the long-tailed duck. Perry et al. [12] observed atypical behaviour in long-tailed ducks which followed a clam fishing boat and fed on different benthic macrofauna prey compared to those ducks that used to utilise the surrounding areas.

A switch in the species’ diet was also recorded in the Baltic Sea, after the invasion of the round goby (*Neogobius melanostomus*). The round goby was found in Lithuanian waters in 2002 and started to reach high population densities after 10 years [27]. The impact of the round goby led to a significant decline in the population density of blue mussels in hard-bottom found in coastal areas. The biomass of the blue mussel was around 2300 ± 1500 g m^−2^ before the round goby’s expansion. Today, the biomass has been assessed to be less than 100 g m^−2^ [27]. As a consequence of the decline in numbers of blue mussels, the long-tailed ducks switched their diet from the blue mussel to other organisms that differed from the living habitat, but also the mobility, size, availability and—most likely—nutritional value, of the blue mussel. This was observed in hard-bottom habitat, while neighbouring soft-bottom habitat remained stable with respect to the status of benthic macrofauna, and hence, the potential contribution to the feeding of long-tailed ducks went unchanged in soft-bottom habitats. On the other hand, the almost complete extinction of blue mussels in hard-bottom habitats made the biomass of the prey organisms comparable between soft- and hard-bottom locations. Large-sized, high-biomass species no longer dominated in hard-bottom habitats, while soft-bottom habitats remained dominated by sub-surface-dwelling bivalve, *Limecola balthica*.

The aim of this study was to evaluate the temporal variation in the diet of the long-tailed duck in habitats that differed by bottom types and the response of the long-tailed duck to the change in the environment. To evaluate these aspects, this study focused on (i) comparing the diet in soft- and hard-bottom habitats, accounting for the sex and the age of the analysed ducks; (ii) evaluating changes in diet composition during the last two decades; (iii) assessing the change in the body condition of the species over time and between different bottom types.

## 2. Materials and Methods

### 2.1. Study Area

The south-eastern Baltic Sea coastal waters of the Lithuanian coast are characterised by different bottom sediments. A relatively uniform distribution of sands is typical along the Curonian spit, whilst the coastal waters along the mainland are characterised by a heterogeneous bottom composed of large boulders, gravels, and pebbles mixed with coarse, medium, and fine sand [28]. Due to these differences, the seabed of the study area was divided into hard- and soft-bottom according to the predominant sediment type (Figure 1).

Macrofauna communities are different in hard- and soft-bottom habitats. The hard-bottom along the mainland coast was dominated by bivalves, *M. edulis trossulus*, which supported a relatively high local diversity of macrofauna (around 25 species), mostly different crustacean and polychaete species [28]. Since 2011, the population of the blue mussel has decreased significantly, and the biomass was reduced from 2300 ± 1500 g m^−2^ to less than 100 g m^−2^ [27] by 2015. This was also followed by a change in the biomass-dominant taxa to barnacles, *A. improvisus,* and a shift from blue mussel colony associated macrofauna to a pool of scarce macrofauna species.

Soft-bottom habitats are dominated by bivalves, *Limecola balthica* or *Cerastoderma glaucum* [28], and some of the species (e.g., *C. glaucum* and *Bathyporea pilosa*) are highly associated with low depths (down to 5–10 m). In contrast to hard-bottom habitats, there was no indication of any obvious shifts in the composition of macrofauna or in its abundance and biomass during the last few decades. Therefore, although the macrofauna biomass is rarely higher than 200–300 g m^−2^, in soft-bottom habitats, it is currently up to 2–3 times higher than in hard-bottom habitats.

### 2.2. Data Collection

The stomach content of 267 long-tailed duck individuals accidently bycaught in fishermen’s gillnets between 2016 and 2020 from November to May was analysed. Before the collection of stomach samples, long-tailed ducks were weighed and dissected for the determination of sex, age, and body index. Stomach content samples were frozen at −20 °C after individuals were removed from the bycatch until the analysis. Before the analysis, stomach samples were defrosted and rinsed with clean water. The prey items were removed and sorted for taxonomic identification to the lowest possible taxonomic unit and for the determination of abundance and biomass [29].

Altogether, 16 stomach samples (6%) did not contain food and were removed from the analysis: 9 of these stomachs were taken from 9 individuals (2 females and 7 males, all adults) caught in hard-bottom, while 7 were from soft-bottom (2 juvenile females, 2 adult female, and 3 adult male individuals) (Table 1).

The ability to count prey individuals depended on the level of digestion. The intact prey items were counted and weighed. Bivalve shells, fish otoliths, and polychaete jaws, were paired and used to estimate the prey abundance as explained in Camphuysen and Leopold [30]. All taxonomically identified prey fragments were weighed and used for characterisation of the biomass of a given prey taxa. The biomass was measured as wet weight with an accuracy of ±0.0001 g after excess water was removed using filter paper. This approach ensured comparability of our results with those obtained in the study of Žydelis and Ruškyte [15].

The body index was estimated by evaluating and summing intestinal fat, subcutaneous fat, and the condition of the pectoral muscle [29]. Fat deposits were scored from 0 (no fat) to 3 (very fat). The same procedure was used to determine the condition of pectoral muscle, where 0 was a strongly emaciated pectoral muscle and 3 was a muscle in good condition [29].

To estimate feeding selectivity (see below), 87 samples of macrofauna data (54 for hard-bottom and 33 for soft-bottom) were collated from various datasets from depths to 25 m (common feeding depth for long-tailed duck [9]) and the time period from 2014 to 2020. For hard-bottom habitats, only benthic samples collected in a range of up to 2–3 km from the bycatch sites were used in the study and considered to represent the particular bottom type, while a larger range of around 13 km was used for soft-bottom habitats due to the much lower spatial heterogeneity of the benthic macrofauna community. Standard Van Veen grabs (size 0.1 m^2^) and steel framed benthic samplers (0.04 m^2^) were used to take samples in soft- and hard-bottom habitats, respectively, with 1–3 replicates in each site.

### 2.3. Statistical Analysis

In the analysis of stomach data, rare taxa that occurred in the diet of less than 2 individuals (<0.8%) were grouped into the higher taxonomic units (Gammaridea, Crustacea, Mysidacea, and fishes) for further analysis or removed from the dataset, as in the case of the Horsehair worm *Godius aquaticus*. Macrophytes were removed from the analysis due to the fact that they had a low occurrence and biomass in the diet.

The Shapiro–Wilk test was used to test the normality of the data. As most of the variables or residuals were not normally distributed and the data transformations did not lead to the normality (or equal variances among the groups), non-parametric tests or robust methods were used. This decision was also driven by the limited accuracy of the diet data on prey diversity, abundance, and biomass due to fact that decomposed prey fragments from stomachs were counted and weighed.

The differences in the diet composition (i.e., biomass or abundance distribution across the prey taxa) of long-tailed ducks between two different bottom types was tested using the Wilcoxon signed rank test. Monthly differences in the prey total biomass, as well as differences in the body index between different bottom types, sex, and age were analysed using the Kruskal–Wallis test. Effects of sex and bottom type on prey diversity, biomass, and abundance were analysed by applying a robust two-way ANOVA for trimmed means [31].

Ivlev’s selectivity index was calculated to assess the long-tailed ducks’ prey preferences in different bottom types using the following equation:(1)$$E=\frac{r_{i}-p_{i}}{r_{i}+p_{i}}$$
where *r_i_* is the proportion of a prey item in the diet; *p_i_* is the proportion of a prey item within the macrofauna community on the seabed; *E* is the selectivity index [32].

All non-parametric statistical analyses were performed using Sigmaplot software (Systat Software Inc., 1735 Technology Drive, Ste 430, Saint Jose, CA, USA). A robust two-way ANOVA for trimmed means (10% trimming level for all tested variables) was implemented employing R WRS2 package according to Mair and Wilcox [31]. The standard error of the mean was used in the study to reflect the accuracy of the calculated average values.

## 3. Results

### 3.1. Seabed Macrofauna Community

In total, 21 macrofauna taxa were present in the hard-bottom habitat, with *Marenzelleria* sp. (85%), *Pygospio elegans* (68%), and oligochaetes (68%) having the highest occurrence. The most abundant were polychaetes, *Fabricia sabella* (Table 2), crustaceans, *Amphibalanus improvisus*, gammarids, and corophiids, while *A. improvisus* (65.1 ± 22.9 g m^−2^) contributed the most to the total biomass (78.4 ± 2.9 g m^−2^).

In the sandy bottom habitat, 26 taxa were recorded and three polychaete taxa *H. diversicolor* (100%), *Marenzelleria* sp. (100%), and *P. elegans* (97%) were almost permanently present. Later, it was found that two taxa were also among the most abundant along with the bivalves *Limecola balthica* and *Mya arenaria*. The total biomass of macrofauna was 285.2 ± 5.1 g m^−2^, with *L. balthica* (123.9 ± 25.5 g m^−2^) contributing the most.

### 3.2. Stomach Data

In total, 33 prey item taxa were found in the stomachs of long-tailed ducks. The average number of prey taxa per individual long-tailed duck was 2.5 ± 0.1.

The taxonomic diversity of prey was significantly different between individuals bycaught in hard-bottom areas (31 taxa) and soft-bottom areas (21 taxa) (Table 3). At the same time, the taxonomic diversity of prey tended to be higher in the stomachs of males compared to females in both types of seabed, although this difference was not significant (Figure 2A). The same pattern is also visible per single individual (Figure 2B), where males were shown to feed on more taxa than females in both bottom types and this resulted in the marginal significance of interaction between sex and substrate (Table 3).

Both, males and females collected from hard-bottom habitats showed higher abundance of prey items compared to those from soft-bottom habitats (Figure 2C), but the differences were not significant (Table 3). The differences in macrofauna biomass contribution to the ducks’ diets in the two seabed types had marginal significance (Table 3; Figure 2D), i.e., macrofauna had a lower biomass in the stomachs of long-tailed ducks from hard-bottom habitats. Although the contribution of fish biomass to the diet was not significantly different between bottom types or individuals of different sex, the biomass of fish prey in the stomachs of males (22.2% and 17%, in hard- and soft-bottom habitats, respectively) was higher compared to that in the stomachs of females (17% and 5.6%, in hard- and soft-bottom habitats, respectively). This largely determined significant differences in total biomass of prey found between males and females (Table 3).

Characterising prey composition, *H. diversicolor* and *A. improvisus* were the most frequent species in the diet of long-tailed ducks (45% and 35%, respectively) in hard-bottom habitats. In soft-bottom habitats, the high prevalence of *H. diversicolor* (61%) in the diet composition was followed by the isopod *Saduria entomon* (47%) and bivalves *L. balthica, C. glaucum*, and *M. arenaria* (13–15%) (Table 4).

Smelt was the most frequent fish taxa in the diet of the long-tailed ducks (18% and 7% in hard- and soft-bottom habitats, respectively), but occurred more frequently in the stomachs of females (42.8%) than in males (31.8%) (Table 4).

#### 3.2.1. Feeding of Juveniles

Seventeen prey taxa were found in the stomach contents of juveniles bycaught in hard-bottom habitats (*n* = 19); the taxa most commonly occurring were gammarids (73%), *H. diversicolor* (7%), and *Idothea* sp. (5%). The occurrence of fish and blue mussels was 2%. In soft-bottom habitats (*n* = 6), *H. diversicolor* (79%) occurred most commonly among the other nine prey taxa and was followed by bivalves (10%) and fish (4%).

Similarly to the diet of adults, the average biomass of prey also differed significantly between the two bottom types: 0.23 ± 0.14 g stomach^−1^ and 0.04 ± 0.02 g stomach^−1^ in hard- and soft-bottom habitats, respectively (Mann–Whitney test, U = 199.5, *p* = 0.019). The average contribution of fish to the total prey biomass in the stomachs of juveniles was equal to 67% and 62% in hard- and soft-bottom habitats, respectively, and considerably exceeded the relative biomass of the other most important prey items, such as crustaceans (21%) in hard-bottom and polychaetes (25%) in soft-bottom habitats.

#### 3.2.2. Monthly Variation in the Diet

No statistical difference between the monthly total biomass of the prey in the stomach content was detected (Kruskal–Wallis test), but fish was the main prey from December to February in the hard-bottom habitat and the diet preference shifted to crustaceans in the other months (Figure 3A). In contrast, in soft-bottom habitats, crustaceans and bivalves remained the preferred prey of the long-tailed ducks during the entire wintering period (Figure 3B).

Ivlev’s selectivity index calculated for hard-bottom habitats demonstrated the highest values for *H. diversicolor* (r_i_ = 0.44, p_i_ = 0.001, E = 0.99) and bivalves (r_i_ = 0.03, p_i_ = 0.002, E = 0.85), while blue mussels were not actively sought from the environment nor avoided (r_i_ = 0.02, p_i_ = 0.02, E = −0.007). For soft-bottom habitats, however, the highest values of the selectivity index were estimated for *S. entomon* (r_i_ = 0.08, p_i_ varied between 2.01 × 10^−5^ to 8.01 × 10^−5^, but E was always equal to 0.99) and *H. diversicolor* (r_i_ = 0.76, p_i_ = 0.06, E = 0.86).

### 3.3. Body Index

No statistical difference was found in the body index between individuals from hard- and soft-bottom habitats (7.2 ± 1.9 and 6.9 ± 1.7, respectively, Mann–Whitney test, U = 12,285, n_soft_ = 101, n_hard_ = 162, *p* = 0.074). Generally, long-tailed ducks were in good condition with a fat score above 5, and only 9–10% of birds had body index values lower than 5. Adults and juveniles had similar body indexes independently of the seabed type; however, body index values lower than 4 were characteristic for adults alone and on average, adult males demonstrated a 6–14% lower fat score than immature individuals.

The analysis results showed that long-tailed ducks with high body index (>5) values corresponded with the higher contribution of crustaceans and *H. diversicolor* biomass to the diet in hard-bottom habitats. In soft-bottom habitats, the higher biomass of *H. diversicolor* only was sufficient for enhanced body index. The results also showed an average 20% decrease in the mean body index from 7.6 ± 0.1 (*n* = 114) to 5.9 ± 0.3 (*n* = 38) between mid- (November–February) and late-winter (March–May) in hard-bottom habitats (Mann–Whitney test, U = 1403, *p* < 0.001), but this was not observed for birds collected in soft-bottom habitats: 6.8 ± 0.2 (*n* = 54) to 7 ± 0.3 (*n* = 38) in mid- and late-winter, respectively (Figure 4).

## 4. Discussion

The results demonstrate that the diet of the long-tailed duck is dependent on the bottom type. Although it is not clear how long the species stays feeding within the same seabed type, prey composition in the stomachs, and particularly that of soft-bodied organisms, generally corresponded to the prey diversity of the seabed type where the long-tailed ducks were bycaught. Next to about 30% higher prey diversity in stomachs bycaught in hard-bottom habitats, more than half of main prey groups (F > 15%, Table 4) were also specific to this bottom type.

The diet of the long-tailed duck is based on the most abundant and available prey in the environment [14], but as it is a small-sized duck with high metabolic rate, it must maximise the energy intake and feed on energetically rich preys. Macrofauna was the preferred prey according to the organism abundance and occurrence in the stomachs of the birds. The diversity of exposed surface-dwelling organisms was obviously higher in hard-bottom habitats, and, therefore, this may explain why more prey items were ingested here compared to soft-bottom habitats. On the other hand, some taxa widely available in the benthic environment were not recorded in the stomach. Small-sized polychaete species, such as *Pygospio elegans* and *Streblospio shrubsolii*, are frequently found in the soft-bottom environment but absent in the diet. This may be related to the low profitability of the species being too small to provide the required amount of energy. At the same time, some mobile species, such as *S. entomon* and *Crangon crangon*, were absent or were scarcely found in the benthic samples but were frequent in the stomachs of long-tailed ducks. Such mismatch is partly related to the low sampling accuracy of these mobile and aggregated benthic species, but also likely is associated with the high feeding selectivity of ducks. Hence, the diet of the long-tailed duck is a result of the complex feeding strategy, which responds to a changing prey taxa diversity, abundance, and biomass in the environment, but also depends on prey distribution pattern, energetic value, and availability (e.g., burrowing depth, seasonal presence due to migration, etc.).

Historically the blue mussel was the main prey of long-tailed ducks in the Baltic Sea (e.g., [14,15]) and its relatively low energetic value among macrofauna [33] was compensated by its high number and the fact that it was easy to access for individuals. However, the availability of blue mussel for the feeding of long-tailed ducks was considerably reduced in the beginning of this century [27]. The blue mussels’ biomass decreased considerably and led to a shift in the diet of the long-tailed duck from blue mussels (which had an occurrence of 92.4% in 1997–2001) to fish (74%) and benthic macrofauna prey (16%) in hard-bottom areas [27]. Our study demonstrated a 20% blue mussel frequency in the stomachs of birds bycaught in hard-bottom habitats. The replacement of blue mussels with other prey was a natural consequence, bearing in mind that the approximate energetic value provided by blue mussels was around 166–391 kJ m^−2^ before its biomass decline and 0.2–0.5 kJ m^−2^ after it. This justifies an extreme change in the energetic source of the main prey when comparing it, for example, to the energetic value of 6.7–9.5 kJ m^−2^ provided by *H. diversicolor* at its actual biomass level of only, approximately, 3 g m^−2^. These figures derived using species average biomass in our study area, species-specific energetic values [32], and available biomass conversion factors [34,35] demonstrate that long-tailed ducks had to compensate for orders of magnitudes of loss of easily available energy resources in hard-bottom habitats by diversifying their diet.

The number of species ingested when feeding in hard-bottom habitats doubled from 17 taxa in 1997–2001 [15] to 31 taxa in 2016–2020 (Table 5). Although to some extent this can be related to the higher taxonomic identification level in our study, the main likely reason behind this finding is a shift to feeding on diverse taxa, such as crustaceans, polychaetes, and fish, as none of them individually have the occurrence and biomass comparable to that of blue mussel before its decline. Additionally, such diversification of prey could be realised at the costs of increased frequency of bird movement between the feeding grounds. This can be justified by the relatively equal biomass and frequency of typical soft-bottom bivalves (*L. balthica*, *M. arenaria,* and *C. glaucum*) in the stomachs of long-tailed ducks caught in both, hard- and soft-bottom habitats. Such high plasticity of long-tailed duck feeding is also supported by other studies. The species demonstrated a preference for Atlantic razor clam, sand shrimp, and isopod in New England [36], while it was shown to mainly feed on *L. balthica* and *M. arenaria* in the south-eastern Baltic Sea [14].

In soft-bottom habitats, the diet of long-tailed ducks changed much less compared to the hard-bottom habitats, and this corresponds well with the fact that there has been a relatively stable benthic macrofauna community in this bottom type during the last two decades. The results show that in soft-bottom habitats, long-tailed ducks foraged less on bivalves and preferred crustaceans, *S. entomon* and polychaetes, *H. diversicolor*, which are more easily accessible or digestible [1] and have similar or higher energetic values compared to the three biomass dominant bivalves (*L. balthica, M. arenaria*, and *C. glaucum*). Although a similar low occurrence of *L. balthica* was noted in the previous study [15], high dominance of *L. balthica* in the diet was found characteristic in the southern Baltic [14]. Clear reasons for these regional differences cannot be traced in our study, but different size frequency distributions of the bivalve populations between sites can be at least partly responsible for this phenomenon.

Our data demonstrate that long-tailed ducks are able to quickly switch their diet from macrofauna taxa to more motile preys, such as fish. In spite of the limited number of analysed juvenile birds, their diet was comparable with the diet of adults. This is in agreement with the results of another study [14] and proves the ability of juveniles to respond to changes in the prey availability as quickly and efficiently as adults. Moreover, such response can be restricted to a certain territory at a relatively small spatial scale, e.g., a distance of a few tenths of kilometres between two different seabed types of our study area.

In this research, the considerable contribution of pelagic smelt in the feeding of long-tailed ducks was observed during the migration period of this fish species from December to February and this was mainly recorded within hard-bottom habitats. After the smelt migration period, long-tailed ducks seemed to return back to feeding on macrofauna organisms, while in soft-bottom habitats the diet of crustaceans was stable during the whole wintering period except for a deviation in December. It remains unclear why Baltic herring (*Clupea harengus membras*) did not become another important fish prey as it is characterised by similar migration outbursts in coastal waters during its spring spawning periods (March–May [37]). This species is highly similar to smelt in terms of individual size, mass behaviour, and exploited habitat, therefore, it may also become a part of the diet of the long-tailed duck in late spring. Although this time period is poorly covered by samples in our study, feeding of the long-tailed ducks on herring eggs (but not on the fish itself) and a good coincidence of bird distribution within herring spawning sites was demonstrated earlier [15,37].

Sea ducks need to forage on energy-rich prey during the pre-migratory period to fatten up for the spring migration and the subsequent breeding season. This is particularly important if one considers scarce food resources in the early breeding season in the Arctic. The presence of good body conditions may affect the breeding success of the species [1] and survival, which have a strong impact on the reproductive potential of long-tailed ducks [13,38]. Generally, our results regarding the body index showed that long-tailed ducks were in good condition and tended to increase their body reserves in mid-winter and decrease before the spring migration in hard-bottom habitats (Figure 4). A similar decline in body reserves was described for other sea ducks in the Baltic Sea [39], which explains this phenomenon as a migratory strategy to minimise fat reserves before performing a non-stop and short-distanced flight to reproductive grounds [6]. Despite the changes in the macrofauna community and the diet composition of long-tailed ducks in hard-bottom habitats, the body index values do not differ significantly from those recorded 20 years ago (Table 5). Moreover, the values are similar between individuals from hard- and soft-bottom habitats indicating that, overall, the species adjusts well to a new feeding condition and the carrying capacity of the seabed seems unchanged with respect to bird feeding at the individual level. However, considerably reduced numbers of long-tailed ducks in coastal waters [27] may also indicate that long-tailed ducks responded to changes in prey availability at the population and not at the individual level. This aspect should certainly be considered in further assessments of the species status in coastal waters of the central Baltic Sea and when forecasting the future scenarios for the species dynamics based on trends in feeding conditions at the seabed.

## 5. Conclusions

This study demonstrated the quick and efficient response of the long-tailed duck (*C. hyemalis*) to extreme changes in the benthic macrofauna community, primarily associated with hard-bottom habitats. A quick shift in the diet was evidence of high flexibility and fast adjustment to new environmental changes; meanwhile, feeding stability in soft-bottom habitats was demonstrated over two decades. Furthermore, the stable value of body index indicated their capacity to feed efficiently and maintain a good body condition, despite the changes in the benthic macrofauna communities. This information has important implications for the conservation of the species, in light of future benthic ecosystem changes that are expected due to climate change. This study may provide support for the development of conservation strategies for the long-tailed duck.

## Figures and Tables

**Figure 1 animals-12-00355-f001:**
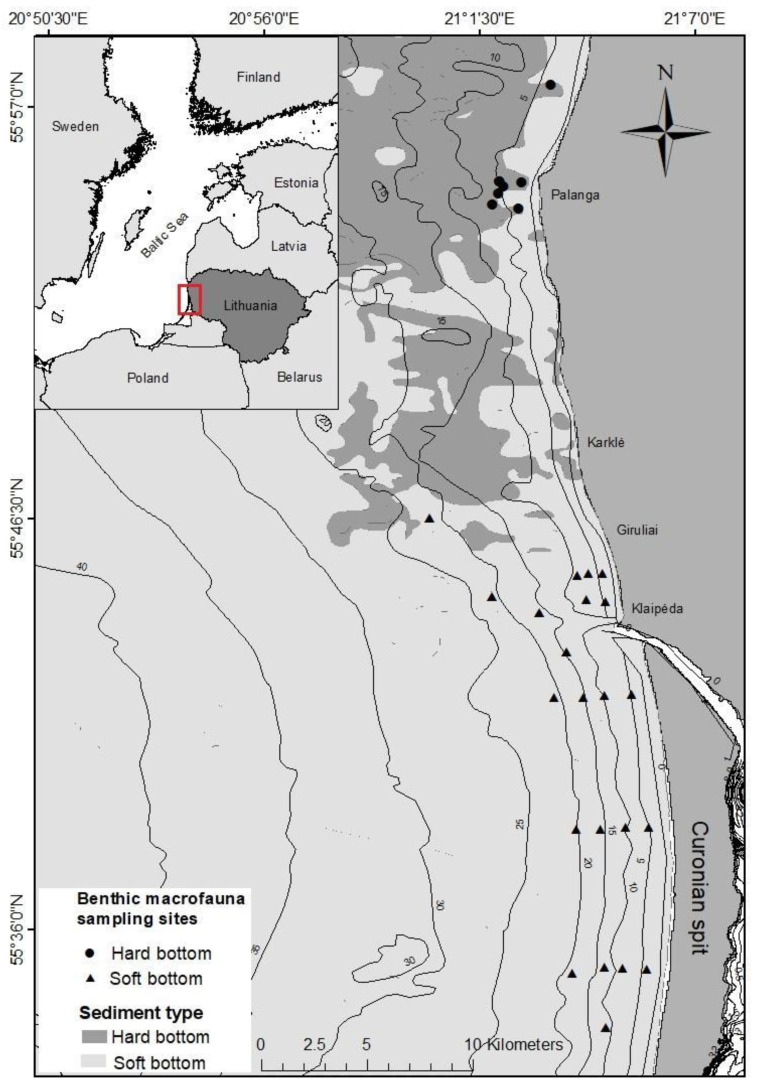
Distribution of hard- and soft-bottom habitats in the study area and locations of macrofauna sampling sites in two bottom types.

**Figure 2 animals-12-00355-f002:**
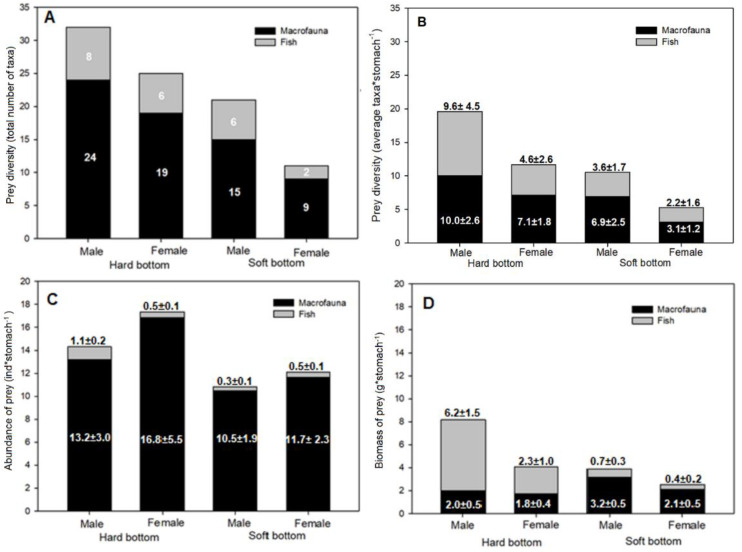
Diversity of prey as (**A**) total number of prey taxa, and (**B**) average number of taxa per stomach; (**C**) average abundance of prey (ind stomach^−1^) and (**D**) average biomass of prey (g stomach^−1^) for two major prey groups of long-tailed ducks collected in hard- and soft-bottom habitats. Labels indicate (**A**) the number of prey taxa, (**B**) the average prey abundance and standard error, and (**C**) the average prey biomass and standard error.

**Figure 3 animals-12-00355-f003:**
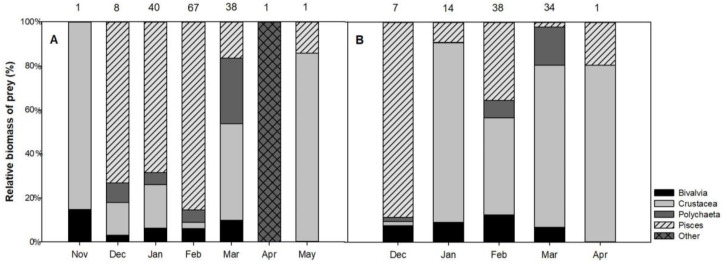
Long-tailed duck’s prey composition according to wet weight biomass (%) during the wintering period in (**A**) hard-bottom and (**B**) soft-bottom habitats. Upper numbers indicate sample sizes.

**Figure 4 animals-12-00355-f004:**
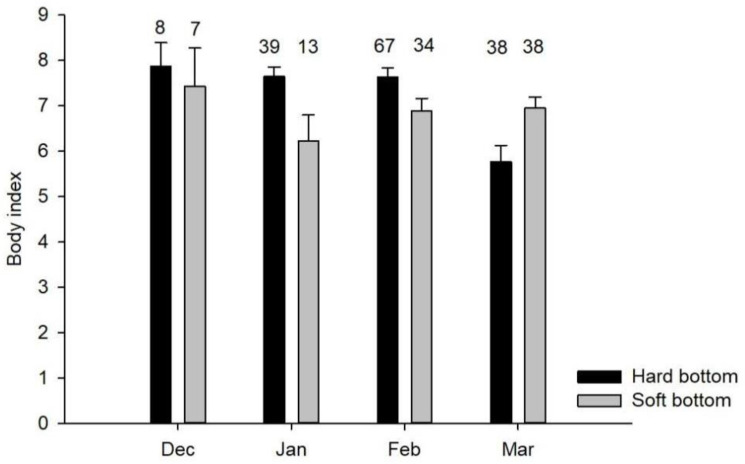
Long-tailed duck body index variation in two bottom types from December to May. Sample size is denoted by labels above histogram columns. Error bars represent the standard error of the mean.

**Table 1 animals-12-00355-t001:** Distribution of analysed long-tailed duck numbers across age groups, sex, and seabed type at the bycatch sites during years from 2016 to 2020.

Soft-Bottom				Hard-Bottom			
Sex	Juvenile	Adult	Total	Sex	Juvenile	Adult	Total
Female	4	27	31	Female	16	42	58
Male	2	62	64	Male	3	95	98
Total	6	89	95	Total	19	137	156

**Table 2 animals-12-00355-t002:** Main characteristics of the macrofauna community in hard- and soft-bottom habitats, according to the data collected from 2014 to 2020. Abbreviations: F (%)—frequency; SE—standard error. Abundance and biomass are expressed in ind m^−2^ and g m^−2^, respectively.

Species/Taxa	Hard-Bottom			Soft-Bottom		
	F (%)	Abundance (SE)	Biomass (SE)	F (%)	Abundance (SE)	Biomass (SE)
*M. edulis trossulus*	51%	280.5 (64.8)	3.0 (2.0)	3%	0.3 (0.3)	<0.1 (<0.1)
*C. glaucum*	19%	2.9 (1.3)	0.2 (0.1)	76%	240.6 (83.3)	67.8 (21.5)
*L. balthica*	49%	20.5 (14.3)	5.5 (3.8)	82%	2080.3 (477.0)	123.9 (25.5)
*M. arenaria*	45%	8.4 (2.4)	0.2 (0.1)	76%	1274.2 (512.1)	32.9 (9.3)
*R. cuneata*				6%	1.2 (0.9)	<0.1 (<0.1)
*A. improvisus*	57%	2584.9 (515.9)	65.1 (22.9)			
*B.pilosa*	9%	0.1 (<0.1)	<0.1 (<0.1)	58%	352.1 (156.3)	0.2 (0.1)
*Corophium* sp.	57%	1056.0 (254.9)	0.4 (0.2)	79%	174.2 (54.6)	0.4 (0.1)
*C. crangon*				3%	0.3 (0.3)	0.4 (0.4)
*D. rathkei*				9%	1.2 (0.7)	<0.1 (<0.1)
Gammaridea undet.	55%	2006.5 (612.7)	0.5 (0.1)	9%	2.4 (1.8)	<0.1 (<0.1)
*Mysidea* undet.				6%	0.6	<0.1 (<0.1)
*I. balthica*	49%	506.2 (181.3)	0.1 (<0.1)			
*J. albifrons*	36%	692.7 (331.1)	<0.1 (<0.1)			
Ostracoda				15%	15.5 (8.8)	<0.1 (<0.1)
*Palaemon elegans*	4%	0.4 (0.2)	<0.1 (<0.1)			
*P. inermis*	6%	0.7 (0.4)	<0.1 (<0.1)			
*Hydrobia* sp.	19%	6.8 (4.3)	0.1 (0.1)	64%	460.9 (152.1)	2.8 (1.4)
Chironomidae				9%	0.9 (0.5)	<0.1 (<0.1)
Nematoda	23%	11.8 (4.2)	<0.1 (<0.1)	48%	70.0 (31.1)	<0.1 (<0.1)
Nemertea				24%	32.4 (12.9)	<0.1 (<0.1)
Oligochaeta	68%	268.4 (144.0)	0.1 (0.1)	88%	1711.5 (394.5)	1.0 (0.2)
*F. sabella*	53%	6261.3 (2748.7)	0.2 (0.1)			
*H. diversicolor*	47%	13.4 (3.7)	2.9 (2.1)	100%	886.1 (170.9)	26.1 (8.3)
*Marenzellaria* sp.	85%	121.7 (27.1)	0.3 (0.2)	100%	2688.5 (429.2)	22.2 (5.9)
*Pygospio elegans*	68%	34.8 (7.0)	<0.1 (<0.1)	97%	4639.4 (1524.2)	1.7 (0.5)
*S. shrubsolii*				73%	428.2 (89.5)	0.2 (0.1)

**Table 3 animals-12-00355-t003:** Numerical outputs (criteria values and *p*-values in brackets) of robust ANOVA with trimmed means for different prey characteristics.

Factors/Parameters	Prey Taxonomic Diversity	Total Biomass	Macrofauna Biomass	Fish Biomass	Total Abundance
Sex	0.0786 (0.780)	3.6285 (0.059)	1.7691 (0.188)	1.5477 (0.219)	1.1519 (0.289)
Substrate	6.1957 (0.015)	0.1046 (0.747)	3.6010 (0.062)	1.7287 (0.194)	0.9362 (0.339)
Sex:Substrate	3.8687 (0.052)	0.1103 (0.741)	1.9272 (0.169)	1.5770 (0.215)	0.0589 (0.810)

**Table 4 animals-12-00355-t004:** Average prey biomass (standard error) and frequency (F%) in the stomach samples of long-tailed ducks collected from 2016 to 2020 in hard- and soft-bottom habitats of coastal waters. Main prey groups (F > 15%) are shown in bold.

Species/Taxa	Hard-Bottom		Soft-Bottom	
	Average Biomass (g)	F (%)	Average Biomass (g)	F (%)
Bivalves				
Bivalvia undet.	0.4 (0.1)	15%	0.5 (0.1)	**21%**
*M. edulis trossulus*	0.7 (0.1)	**20%**	0.9 (0.1)	6%
*L. balthica*	0.4 (<0.1)	8%	0.3 (<0.1)	14%
*C. glaucum*	0.5 (0.1)	3%	0.2 (0.1)	13%
*M. arenaria*	0.9 (0.1)	**19%**	0.3 (0.1)	15%
Crustacea				
*G. zaddachi*	0.5 (0.1)	1%		
Gammaridae undet.	1.3 (0.2)	6%	<0.1	1%
*D. villosus*	0.3 (<0.1)	2%		
*S. entomon*	1.3 (0.2)	**22%**	4.5 (0.4)	**47%**
*N. integer*	1.2 (0.2)	9%		
*Idothea* sp.	0.1 (<0.1)	1%		
*A. improvisus*	0.4 (0.1)	**35%**	0.4 (0.1)	11%
*P. elegans*	<0.1 (<0.1)	1%		
*C. crangon*	1.6 (0.1)	8%	1.8 (0.3)	4%
Crustacea undet.	0.3 (<0.1)	4%		
*P. lacustris*	<0.1	1%		
Mysidae undet.	<0.1	1%	<0.1 (<0.1)	1%
*Corophium* undet.	<0.1	1%	<0.1	1%
Polychaeta				
*Marenzelleria* sp.	1.2 (0.1)	1%		
*H. diversicolor*	1.3 (0.4)	**45%**	0.6 (0.1)	**61%**
Fishes				
*O. eperlanus*	23.3 (1.6)	**18%**	6.1 (0.5)	7%
*A. tobianus*	0.2 (<0.1)	1%	2.1 (<0.1)	2%
*P. minutus*	0.7	1%		
*P. flesus*	11 (0.7)	3%	2.1	1%
*G. aculeatus*	0.2 (<0.1)	2%	<0.1	1%
*N. melanostomus*	<0.1	0%	<0.1	1%
*M. scorpius*	1.0	1%		
*G. cernuus*	3.2	1%		
*C. harengus*	<0.1	1%		
*Pisces* (undetermined)	2 (0.3)	**20%**	0.7(0.1)	**18%**
Others				
Gastropoda undet.	<0.1 (<0.1)	1%	<0.1 (<0.1)	4%
Bottom macrophytes	<0.1 (<0.1)	13%	<0.1 (<0.1)	6%
*G. aquaticus*	<0.1	1%		

**Table 5 animals-12-00355-t005:** Comparison of the long-tailed duck diet characteristics analysed in this study and Žydelis and Ruškyte (2005). Bottom type divided in hard-bottom (H) and soft-bottom (S).

Characteristics	Bottom Type	Žydelis and Ruškyte, 2005	This Study, 2022
Number of stomach samples per bottom type (*n*)	H	119	156
	S	89	95
Abundance of prey items in stomachs	H	2.2 ± 1.1	5.8 ± 1.1
	S	1.9 ± 1.2	4.7 ± 0.7
Number of prey items found in stomachs	H	17	31
	S	18	21
Frequency (%) of main prey in stomachs:			
*Mytilus edulis trossulus*	H	92.4%	20%
*Limecola balthica*	S	16.1%	14%
*Mya arenaria*	S	17.2%	15%
*Saduria entomon*	H	1.7%	22%
	S	71.3%	47%
*Osmerus eperlanus*	H	0%	18%
	S	1.2%	7%
*Hediste diversicolor*	H	0%	45%
	S	14.9%	61%
Ivlev’s selectivity index:			
*Mytilus edulis trossulus*	H	E = −0.05	E = −0.007
*Saduria entomon*	S	E = 0.73	E = 0.99
Occurrence of macrofauna in stomachs		62%	63.8%
Body index	H	6.9 ± 1.9	7.2 ± 1.9
	S	7.4 ± 1.4	6.9 ± 1.7

## Data Availability

The stomach data (biomass and abundance) and body scores are available from Dryad repository (https://datadryad.org), https://doi.org/10.5061/dryad.fxpnvx0th.

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
