# Peer review of "Response of Long-Tailed Duck (Clangula hyemalis) to the Change in the Main Prey Availability in Its Baltic Wintering Ground"

_animals, 2022, doi:10.3390/ani12030355_

Round 1

Reviewer 1 Report

Generally manuscript is interesting and provide a new data. Shift in a diet of long-tailed duck has been found in the studied area in comparison to the previous study 20 years ago. It is interesting that fish and crustaceans were dominant prey (percentage by wet mass) of birds in all studied months in both types of bottom. Despite the change in food, birds showed a good condition (body index) in hard and soft bottoms. The data was presented clearly but I found some mistakes and incompatibilities especially in methods.

A short history of the round goby expansion in the Baltic sea and the impact of this species on the ecosystem should be given in introduction. I suggest to use term “macrobenthos” instead “macrofauna”. Number of macrobenthos samples was small. Only 33 sample in the soft bottom between 2014 and 2020 was collected. Samples of macrobenthos was described too poorly. What was water depth of sampling? When (month) benthic samples were collected? How many samples was taken from one place? Details of fish catches should be added also. It is not described in the methods how biomass of particular taxa was measured. Fish otoliths should be paired and measured to estimate fish length based on the regression formula. Estimated fish length are used to calculate fish weight based on the regression formula. Similar size of mollusks shells should be measured to estimate their length and biomass. This also applies to the estimation of the biomass of other taxa. The method of estimating the biomass of other taxa should also be given.

Comparison of the macrobenthos in habitat and diet showed unclear results. Hediste diversicolar is frequent in the diet and macrobenthos community in the soft bottom. However many frequent species or with high abundance or biomass in the macrobenthos community were not significant in the diet of long-tailed duck. It is strange that Saduria entomon was important prey in a diet (frequency and biomass) but was not recorded in the macrobenthos community in the soft bottom. Similar Crangon crangon was found in the diet but it was not recorded in the macrofauna community in the hard bottom. This phenomena should be explained and differences should be more described in discussion. Moreover why Limecola (Macoma) baltica reaching high biomass in the habitat was not important prey of long-tailed duck foraging in the soft bottom? This species was an important component of the long-tailed duck’s diet in other studies from the Baltic Sea.

Latin name in the title should be written in italic.

Page 4, row 12 from bottom – Give months of drowned birds: "from November to May".

Page 4, row 6 from bottom - Better give that “Altogether 16 stomach samples (6%) were not contained food”.

Page 4, row 3 from bottom - Semicolon is redundant after “3 adult male individuals”.

Page 6, row 2 from upper - Frequency of Pygospio elegans was given 68% in Table 2 and in the text is 72%.

Page 7, row 6 from bottom – Frequency of H. diversicolor is 45%, not 47%. Amphibalanus improvises should be added because its frequency is 35%, greater than bivalves.

Page 8, row 6-3 from bottom - This can be coincidental and weak relationship and it is unconvincing.

Page 10, row 4 from bottom – In the text is “to 5.9 ± 0.3 (n= 40)” and on the figure 5 is 5.9 and n= 38.

Page 10, row 2 from bottom – In the text is “to 7 ± 0.3 (n=39)” and on the figure 4 is 7.0 and n= 38.

Legend of Figure 4 does not fit on the page.

Discussion is interesting and well written. References in many places were not in accordance to the guidelines for authors. For example after surname should be given semicolon and before range of pages should be given colon. Shortcut pp. should be given before pages.

Reviewer 2 Report

Comments to the Authors 

This interesting paper describing winter diet and its changes in one of the duck species occurring during winter on the Baltic Sea. Data presented in this manuscript are rare and I think are worth of publication. In general, the manuscript is ok, but I found some issues which should be addressed by authors (below details).

General comments:

  1. The last paragraph of the introduction is quite enigmatic. What is your main question to which you want to answer? I would expect in this part that you present a clear aim of this study and maybe propose some predictions regarding to expected results. Please, clarify it.

  1. Your data on juvenile individuals are very small. Such sample size is not proper to statistical testing (e.g. to compare adult and juvenile individuals). Furthermore, there is a huge bias in sample sizes between two age classes. I realise that you used non-parametric test which, to some extent, allow to analyse small sample sizes, but still results of such test can be by-product of random processes rather than reliable differences. So, my recommendation is to focus in this paper on adult data only. Of course, you can present some descriptive statistics on juvenile birds regarding to their diet, but you cannot use statistical test to show differences between groups due to small number of observations and between-group sample bias.

  1. You used non-parametric tests, why? Parametric tests have higher power and conclusions based on them are stronger. You pointed that you tested for normality distribution of data. It means that all data (dependent variables) had skewed distribution and this is reason why you performed non-parametric tests? This should be mentioned in the text. Anyway, still you can perform proper data transformation and use parametric test, like t-test or Anova. Why didn't you do it?

  1. I think that written English used in the paper needs improvement by native speaker.

  1. I suggest avoid in the text of some emotional words, like “dramatic”, “drastic” because this scientific paper, and replace them e.g. words “large” or “extreme”.

Specific minor comments:

Title: I think that the title should be shortened, to me is too long and sounds too much dramatic, which do not represent actual content of the paper. My proposition is: “A Response of Long-Tailed Duck (Clangula Hyemalis) To the Change in the Main Prey Availability on Its Baltic Wintering Grounds”.

Last paragraph of the introduction: The sentence “we analyse the long-tailed duck adaptation to the changed environment” is not true. I am not convinced that you truly studied in this paper any “adaptation”. You simply examined diet of the duck species in relation to habitat, sex, a period of time etc. Note, that adaptation is a trait which increases survival and reproduction of individuals. You do not examine none of them and in fact you do not know if a change in diet affects those fitness components. Moreover, a shift in diet do not have to result from adaptation to a new environmental conditions but be caused by the lack of other preys (especially as a new prey displaced old one). So, you should avoid using this term in this context.

Figure 1.: The quality and clarity of the figure is poor. You should correct it.

3.2.1. Feeding of juveniles: Results of this section are not reliable for me due to small sample size and sample biases between groups (see General comments, point 2).

Discussion: “Our data demonstrate, that under favourable conditions”: what does mean “under favourable conditions”.

Discussion: “The diet of juveniles was comparable with the diet of adults, what is in agreement with other study [14] and proves ability of juveniles to adapt to the changes in the environment as fast and efficient as adults.” How I said data on juveniles are not enough trustable. Second, your data do not prove any adaptation, you only showed ability to shift or adjust diet in response to changing environment. It should be changed over the all text of the manuscript, because you do not examine any adaptation here. You should be accurate in using words.

Reviewer 3 Report

The authors compare the diet of ducks caught on hard and soft bottom habitats. Generally, the study is interesting and timely considering the changes currently happing in the Baltic Sea. Also, the population of the long-tailed duck has shown a marked decrease so this is an import study in trying to identify the reasons for that decline. The text is quite easy to follow but the English would need major editing, preferably by a native speaker. Especially the way sentences are constructed is often problematic for the understandability of the text.

The main shortcoming of the paper is the data analysis. The authors use conventional non-parametric methods, which may not be adequate for the sampling design. The sampling design calls for a factorial design analysis that could properly separate the effects of habitat, sex and age of the birds.

Introduction

Introduction of bycatch by gillnets comes a little surprisingly as the reader does not expect such terminology about birds. Perhaps there could be an introductory statement about “main causes for decline”.

Methods

End of page 3: You say that the biomass on soft bottoms is an order of magnitude higher than on hard bottoms, but report 200-300 m-1 and 100 g-1 for soft and hard bottoms respectively. These number do not represent an order of magnitude difference. Clarify.

Second last paragraph before 2.3. What is meant by “near” in the sentence ” Only samples collected near the bycatch areas…”? Provide more precise measure (m, km?).

It is not clear why you use smelt data? Provide a motivation for this choice and tell how it relates to the present study question.

Results

What is the variable measuring diet composition? In the section on Stomach data you compare (Wilcoxon) this between the bottom types but you do not explain how this variable is obtained in the methods.

I wonder about the analysis of the data in Figure 2. You say you use a Kruskal-Wallis but this would indicate that you are comparing four independent samples (also indicated by df’s). However, in reality your setup is more complex and follows a 2x2 factorial design where bottom type is one factor and sex is the other. This should be analysed using a generalized linear model that allows you to test for the independent effects of sex and bottom type and their interaction. Now you say that you had now significant differences between sexes but you do not provide any statistical support for this which a K-W cannot provide.

3.2.1. Report statistical evidence for a lack of significance between juveniles and adults.

Figure 4: What is on the y-axis?

3.2.2. How is the effect of month tested? Report.

3.3. Why is body index not analysed separately for males and females?

Figure 5: What do the error bars stand for?

Discussion: You say that stomach content corresponded to prey availability in the different habitats. However, you provide no explicit test of this in the results, so it is up to the reader whether to believe you or not. It would be important to have some kind of quantification of this relationship.

“Macrofauna was the preferred prey according to the organism abundance and occurrence in the stomachs of the birds” Did you assess the use of any other types of prey than macrofauna?

Round 2

Reviewer 1 Report

The manuscript titled “Quick response of long-tailed duck (Clangula hyemalis) to the dramatic change in the main prey availability in the wintering ground of the south-eastern Baltic Sea” was improved and now it is better to read. Authors introduced some corrections to the introduction and to the discussion. Some changes were made in results also. However more explanations and corrections are needed.

In the manuscript is given that “The intact prey items were counted and weighted. Prey fragments, such as pieces of mollusc shells, fish otoliths and polychaete jaws, were paired and used to estimate the preys’ abundance and the biomass of a given prey taxa”. Authors informed that the precise estimation of weight of each taxa was not possible especially for most of the soft-bodied benthic organisms. This raises doubts as to the methodology used. Biomass estimation is possible based on numbers and average weight of the particular taxa. Remains of prey found in the stomachs and the diagnostic features of particular taxa give possibility to determine prey numbers. It is probably impossible to do in this study. The method should be given clearly that only undigested preys found in stomachs were weighted. It does not allow an estimate of the real biomass in the stomachs. Digestion time is different of particular preys. This may affect the likelihood of finding undigested preys in the stomachs. So this technique probably allows an estimate of approximate biomass only.

Authors do not think as necessary to give information that frequency of Amphibalanus improvisus  in the diet was 35% from hard bottom because frequency of this species was only 11% from soft bottom. However they give that Mytilus edulis trossulus was the most frequent species in the diet (20%) for the samples from hard bottom and the frequency of this species from soft bottom was only 6%. Moreover frequency of Saduria entomon in a diet was 22%, more than Mytilus edulis trossulus.

The above comments should be clarified and corrected before the publication of the manuscript.

Reviewer 2 Report

Comments to authors

I reviewed the revised version of the manuscript and I find the new version clearly improved. I appreciate the authors answered all my questions and reacted to my comments. I have no further questions or comments. I am convinced the manuscript can be published.

Author Response

We thank the reviewer for his helpful comments to improve our manuscript.

Reviewer 3 Report

I thank the authors for the revision, and I am mostly happy with the corrections. Regarding the trimmed means it would be important to report how much trimming you did and whether the same trimming was applied to each variable for which the ANOVA was ran? Another approach would have been to apply a generalized linear model (or similar) for which you cold choose the appropriate probability distribution.

A few points about your formulations of the results text:

“Long-tailed ducks were shown to have a significantly different diet composition in the two bottom types (Table 3), and the taxonomic diversity of prey was higher for individuals bycaught in hard-bottom areas (31 taxa) compared to soft-bottom areas…” This should be simplified to read “The taxonomic diversity of prey was higher for individuals bycaught in hard-bottom areas (31 taxa) compared to soft-bottom areas (Table 3)…”

“The differences in importance of macrofauna biomass contribution to the ducks’ diet in the two seabed types had marginal significance (Table 3; Figure 2d)” In the results section you should not interpret your results, just report them. Rewrite “The differences in macrofauna biomass contribution to the ducks’ diet in the two seabed types had marginal significance (Table 3; Figure 2d)”
